# System for Non-Contact and Multispectral Examination of Blood Supply to Cutaneous Tissue

**Michal Labuda ***, **Maros Smondrk**, **Branko Babusiak** and **Stefan Borik**

Department of Electromagnetic and Biomedical Engineering, Faculty of Electrical Engineering and Information Technology, University of Zilina, 010 26 Zilina, Slovakia
* Correspondence: michal.labuda@feit.uniza.sk; Tel.: +421-41-513-5097

**Abstract:** The presented system for non-contact examination of tissue perfusion is one of the tools for complex examination of human body tissues (skin, subcutaneous) and their mutual interactions, including blood flow and activity under various external stimuli. In our system, optical radiation, with wavelengths of 525 nm, 625 nm, and 940 nm, was used to investigate the perfusion and properties of skin tissue. Our work presents that it is possible to obtain comprehensive information about the cardiovascular system and skin tissue perfusion by a suitable combination of wavelengths, light intensity, and homogeneous illumination distribution with a properly chosen sensing device—a camera. The proposed system consists of an illumination device that emits light of the three wavelengths mentioned above and thus makes it possible to investigate the skin tissue structures and their interrelationships in terms of their blood supply and interactions with each other.

**Keywords:** cutaneous tissue; illumination device; multispectral PPGI; perfusion; photoplethysmography imaging

## 1. Introduction

*Cutaneous tissue* is a complex system composed of multiple layers that are interconnected and contain different compartments [1]. For example, part of the skin is traversed by blood vessels [2], which are also connected to deeper tissue structures [3], which include muscle tissue. Due to the large muscle mass, approximately 25% of cardiac output flows through cutaneous tissue at rest [4]. Therefore, disease-induced changes in this vasculature may contribute significantly to the pathological consequences of systemic disorders [4]. For example, disturbances in the perfusion of skin and muscle tissue can cause diseases such as the development of compartment syndrome [5], chronic venous insufficiency [6], and muscular dystrophy [7]. In addition, pathological changes in the vasculature of skeletal muscle can occur in hypertension [8], hypercholesterolemia [9], obesity [10], diabetes [11], aging, and chronic portal hypertension [12]. Therefore, understanding the function of the cardiovascular and neuromuscular systems and their interconnections is necessary to identify and initiate treatment for each disease [4].

The simultaneous use of an electrocardiogram (ECG) and a photoplethysmogram (PPG), while keeping a distance separating the site of the ejection of the systolic pulse and the site of measuring the PPG signal, can help to determine the pulse wave velocity (PWV) [13]. Optical coherence tomography (OCT) and the method employing the photoacoustic effect (PAT) are also among the methods that use light as a source of information about the patient's state of health. OCT is a noninvasive high-resolution optical imaging technology based on interference between the signal from the object under investigation and a local reference signal [14]. PAT is cross-sectional or three-dimensional imaging based on the photoacoustic effect. PAT combines high ultrasonic resolution and strong optical contrast in a single modality, capable of providing high-resolution structural, functional, and molecular in vivo imaging in optically scattering biological tissue at new depths [15].

In our work, we focused on the novel application of PPG and its non-contact unobtrusive form [16–18]. An illumination source and a sensing device, a camera, are used as the standards for the non-contact examination of subcutaneous blood flow [19]. In addition, white light generated by light-emitting diode (LED) technology or monochromatic light is most often used as the light source [20].

As an innovative method of non-contact investigation of cutaneous perfusion in different tissue layers, we can consider the use of light with selected wavelengths. The use of multiple wavelengths affects the depth of light penetration into the tissue. At the same time, this approach can provide a better picture of what is happening in different cutaneous layers in terms of perfusion in different parts of the arterial tree. Such a technical solution can, e.g., examine the blood supply to the cutaneous tissue and its adjacent layers.

There are several scenarios in the literature where authors have used an illumination source with different wavelengths. In [21], they used a system combining the laser Doppler flow method (LDF), working with a HeNe laser with a wavelength of 632.8 nm, and non-contact PPG based on 560 nm and 810 nm. This study revealed that using several wavelengths is important in terms of their penetration to different layers of the tissue, allowing for their individual investigation. In [22,23], they studied the rhythmic phenomena in skin perfusion using a near-infrared camera. Blazek and Blazek, in [24], described the evaluation of dermal blood perfusion before and after standardized clinical skin tests were performed under clinical control in the allergology field. This study supported this qualitative evaluation of skin blood perfusion with PPG, using a green light as a monochromatic illumination source. In addition, this study used LED near-infrared light with wavelengths of about 940 nm to reach deeper skin layers. Other authors [25] described the non-contact imaging of arterial oxygen saturation ($SpO_2$) distribution in tissue based on detecting a two-dimensional matrix of spatially resolved PPG signals at different wavelengths. As a first step towards $SpO_2$ imaging, they built a device using a monochrome CMOS camera with an apochromatic lens and a 3-$\lambda$ illumination device ($\lambda_1$ = 660 nm, $\lambda_2$ = 810 nm, $\lambda_3$ = 940 nm; a total of 100 LEDs).

Our work provides an alternative and affordable solution combining microcontroller-controlled LED panels, which allow for the simultaneous sensing of the PPGI signal with three wavelengths using a machine vision camera, and possibly another external reference in the form of a selected biosignal, according to the desired experiment scenario.

## 2. Materials and Methods

### 2.1. System Functional Description

The proposed system consists of a total of eight blocks (Figure 1). The main element of the whole system represents a microcontroller (MCU), which controls the illumination device and the camera. The MCU also determines the exact time to capture the image, so it ensures synchronization between the illumination device and the camera. Thus, the image must be captured at a specific time when the radiation of the desired wavelength is emitted. The rising edge of the control pulse–width modulation (PWM) pulse from the MCU marks the beginning of the camera's image capture. The captured image is then transferred to a computer via a USB 3.1 interface.

The entire designed system is powered by a DC switching power supply with an output voltage of 24 VDC. A keyboard is used to enter the system inputs. The individual inputs are used for the user configuration of the system. The display can show the current measurement status, including the set system parameters. The display is controlled by the MCU via the I2C communication interface.

The illumination part of the device was designed using a precisely defined number of LEDs. The proposed device uses three wavelengths (525 nm, 625 nm, and 940 nm), so the proposed device can be referred to as "3-$\lambda$". The device also allows for controlling the switching frequency between the individual illumination blocks with different wavelengths and for setting the number of functional standalone blocks formed by the LEDs. The illumination device can work in a sequential pulse mode or a continuous mode. The device

also contains a diffusion filter, which ensures a homogeneous distribution of the radiation in the entire beam angle of the illumination device. Ensuring the homogeneity of the irradiation is crucial in terms of eliminating optical artifacts during the measurement.

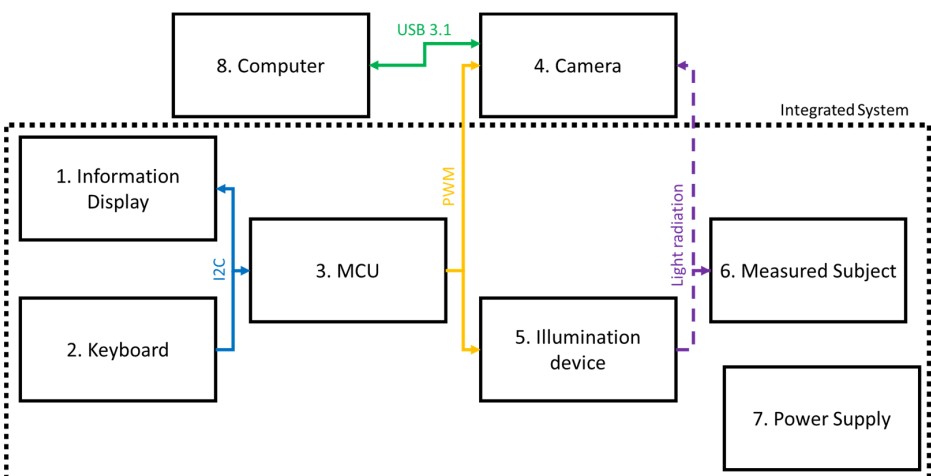

**Figure 1.** Block diagram of the proposed system.

The images are recorded in the ".raw" format, which ensures that the measured data will not be modified by another method of lossy compression. Finally, individual images are processed and analyzed in a tailor-made application on a computer. The application was designed in MATLAB (Mathworks, Portola Valley, CA, USA).

*2.2. System Operation Based on Timing Diagram*

The central control unit of the whole device is the ATmega328P (Microchip Technology, Chandler, AZ, USA) MCU, which is part of the Arduino Uno development board (Arduino, Ivrea, Italy). The MCU sends a control pulse to the sensing unit—the camera, which responds by taking a picture when the control pulse is received. The timing diagram of the proposed system is shown in Figure 2. The total image-taking cycle can be divided into 12 equal intervals, $C_i$. The length of each interval differs depending on the selected frame rate at the device's initial setup. The mentioned intervals were created for a clearer and more practical timing of the proposed system. The selected task is performed at each interval (e.g., turning on an LED with a specific wavelength or sending a control pulse to the camera).

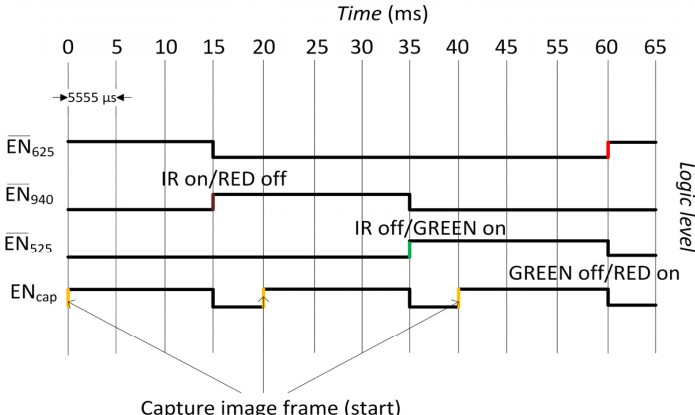

**Figure 2.** Timing diagram of the proposed system, rising edges of enable signals $\overline{EN}_{525}$, $\overline{EN}_{625}$, $\overline{EN}_{940}$, for LEDs with wavelength of 525 nm, 625 nm and 940 nm are highlighted with green, red and brown color. Rising edge of capture frame signal $EN_{cap}$ is highlighted with yellow color.

In this case, we chose the frame rate $F_s$ = 15 Hz per channel, and one interval lasts

$$T = \frac{1}{C_i \times F_s} = \frac{1}{12\ \text{Hz} \times 15\ \text{Hz}} = 5555\ \mu s \tag{1}$$

The selected number of intervals was tested as the most suitable for the system reliability and stability. If we look at the timing diagram in Figure 2, it describes the 3-λ mode and the sequential pulse mode. An important part of the control signal $EN_{cap}$ for the camera is the rising edge when the image is captured. The rising edge of the control signal is always brought to the camera synchronously based on how the individual LEDs of the corresponding wavelengths are activated. Subsequently, it takes time for the camera to take an image. Therefore, we had to reserve extra time for the camera to maintain the system's correct operation. During the image processing, the proposed system also activates a timer for the next control pulse in case of a sampling frequency of 15 Hz per channel, according to:

$$F_s = \frac{1}{C_i T_t} = 15\ \text{Hz} \tag{2}$$

### 2.3. Adjustability of Illumination Device

If we look at Figure 3, we can see that the adjustability of the illumination device can be divided into four main blocks. The first and second blocks relate to the light mode of the lighting device, and thus, to the setting of the 1-λ, 2-λ, or 3-λ mode and the operating mode, that is, the continuous or switching mode (repeated switching between different wavelengths). The third block defines the number of active standalone working units; this option can increase the illumination power of the illumination device. The fourth block defines the sampling frequency of the system.

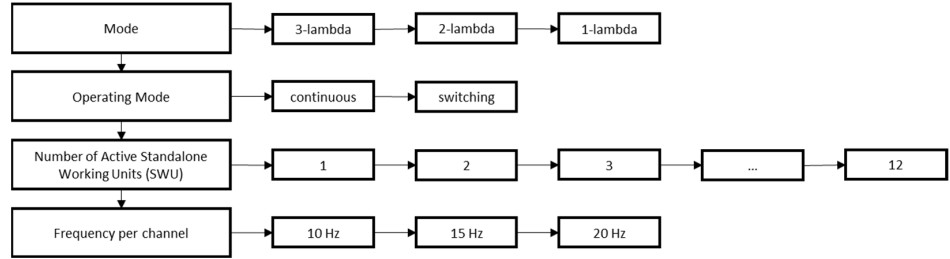

**Figure 3.** Adjustability of illumination device.

At the system initialization, it is important to select the operating mode of the illumination device (see Figure 3). If only surface, subsurface, or deep structures (cutaneous, subcutaneous tissue) are examined, the 1-λ mode is sufficient. However, the combined 2-λ and 3-λ modes should provide a more detailed picture of the perfusion relationships between the individual tissue structures. The next step is to select the operating mode. Here, it is possible to select the continuous and sequential pulse (switching) modes. The continuous mode is feasible and makes sense only in the 1-λ mode when examining the tissue structures only in one wavelength. On the other hand, the sequential pulse mode can be described as switching between individual wavelengths with a predefined frequency. This mode allows for the use of the 2-λ and 3-λ modes to investigate multiple structures (tissues) simultaneously and under the same measurement conditions. We can call it simultaneous because of the frequency band of the measured signal. The penultimate option is to choose the number of active functional units or standalone working units (SWU), as shown in the adjustability diagram of the illumination device in Figure 3.

The functional standalone working unit (SWU) consists of five LEDs for each wavelength, for a total of 15 LEDs (Figure 4). In Figure 4, the green color corresponds to 525 nm LEDs, the red color corresponds to 625 nm LEDs, and the brown color corresponds to

940 nm LEDs. The LED driver is marked as blue. The technical parameters of the chosen diodes are shown in Table 1.

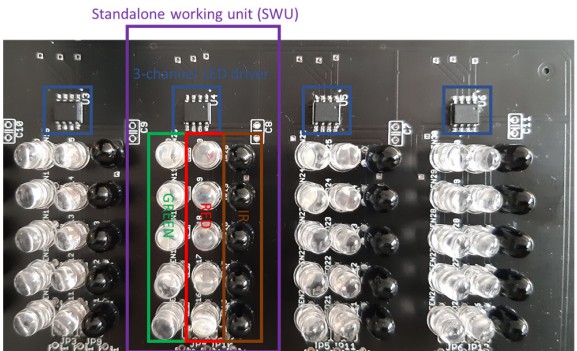

**Figure 4.** Standalone working unit of the illumination device.

**Table 1.** Selected technical parameters of used LED diodes [26–28].

| Type | Wavelength (nm) | $I_F$ (mA) | $V_F$ (V) | Radiance (mW/sr) | Half Angle (°) |
|---|---|---|---|---|---|
| OSG59A5111P *[1] | 525 (GREEN) | 50 | 3.1 | 260.8 | 8 |
| OS6YKA5201P *[2] | 625 (RED) | 75 | 2.3 | 543.9 | 8 |
| SFH4544 *[3] | 940 (IR) | 100 | 1.6 | 550 | 10 |

*[1,2] (Optosupply, Hong Kong), *[3] (Osram, München, Germany).

The illumination device allows for selecting a minimum of 1 active SWU and a maximum of 12 units that create the device (Figure 5). The last adjustable part of the device is the configuration of the frequency of image capture by the camera (sampling frequency), and the following options are offered: 10 Hz, 15 Hz, or 20 Hz per channel. When operating the device in 2-λ or 3-λ in the sequential pulse mode, the microcontroller ensures the synchronicity of the camera and the illumination device.

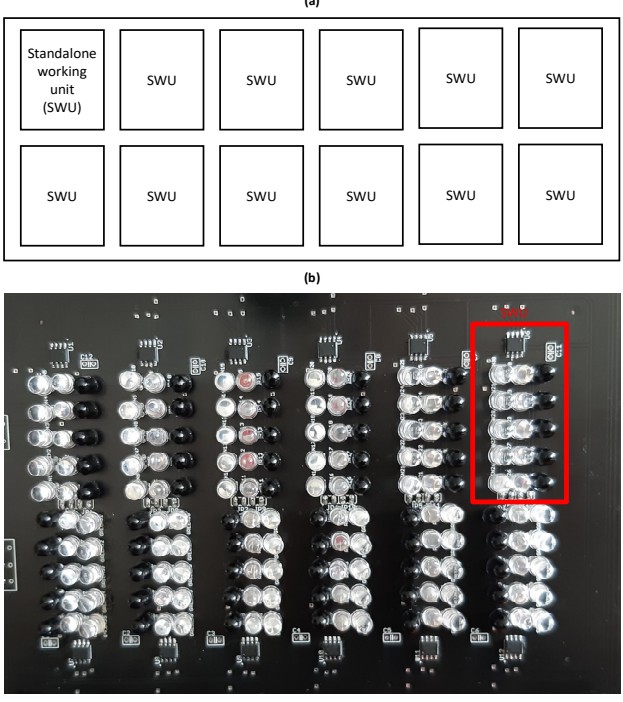

**Figure 5.** PCB layout of the illumination device: (**a**) simplified; (**b**) physical layout.

The device is powered from a switched mode DC power supply, AMES100-24S277NZ (AIMTEC a. s., Pilsen, Czech Republic), with a voltage output of 24 VDC and a maximum guaranteed power of 100 W, Table 2.

**Table 2.** Selected technical parameters of the power supply [29].

| Type of Power Supply Used | AMES100-24S277NZ 24V |
|---|---|
| Recommended AC input voltage | 85 V–305 V |
| DC output voltage | 24 V |
| Output current | 4.5 A |

When designing the hardware part of the illumination device, it was necessary to consider the time and voltage stability of the device. Therefore, the voltage source mentioned above was chosen. Simultaneously, the operation of this voltage source was transformed into a current source (CL320—3 Channel LED driver). The LEDs are therefore controlled by a current LED driver, which, in the initial tests, showed the best stability required for the operation of our device. The LED diodes for each wavelength are connected in a series and create LED strings (Figure 6). Subsequently, we calculated the electrical parameters of the device:

$$V_{cc_{625}} = V_{F_{625}} \cdot N = 2.8 \text{ V} \cdot 5 = 14 \text{ V}$$
$$I_{c_{625}} = I_{F_{625}} \cdot N_{SWU} = 0.020 \text{ A} \cdot 12 = 0.24 \text{ A}$$
$$P_{c_{625}} = I_{c_{625}} \cdot V_{cc_{625}} = 0.24 \text{ A} \cdot 14 \text{ V} = 3.36 \text{ W}$$
$$V_{cc_{525}} = V_{F_{525}} \cdot N = 3.1 \text{ V} \cdot 5 = 15.5 \text{ V}$$
$$I_{c_{525}} = I_{F_{525}} \cdot N_{SWU} = 0.020 \text{ A} \cdot 12 = 0.24 \text{ A}$$
$$P_{c_{525}} = I_{c_{525}} \cdot V_{cc_{525}} = 0.24 \text{ A} \cdot 15.5 \text{ V} = 3.72 \text{ W}$$
$$V_{cc_{940}} = V_{F_{940}} \cdot N = 1.6 \text{ V} \cdot 5 = 8 \text{ V}$$
$$I_{c_{940}} = I_{F_{940}} \cdot N_{SWU} = 0.020 \text{ A} \cdot 12 = 0.24 \text{ A}$$
$$P_{c_{940}} = I_{c_{940}} \cdot V_{cc_{940}} = 0.24 \text{ A} \cdot 8 \text{ V} = 1.92 \text{ W}$$

(3)

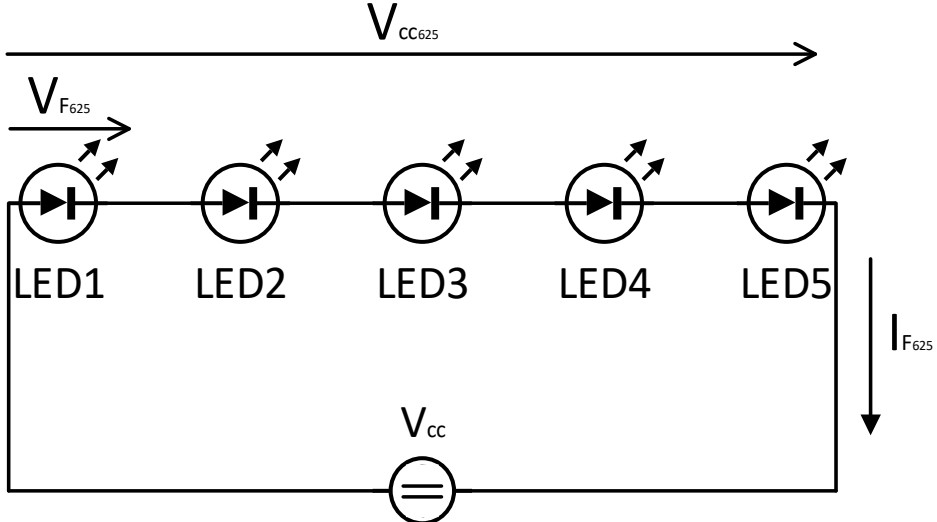

**Figure 6.** Connection of one LED string.

### 2.4. Three-Channel Linear LED Driver

Since we chose the current-regulated LEDs connected in a series because they exhibited higher stability and fewer transients than their parallel alternative with voltage regulation, we had to select a suitable constant-current LED driver for the selected voltage source. We

used the CL320 integrated circuit (Microchip Technology, Chandler, AZ, USA) designed to control up to three LED strings (3 channels) at a constant electric current of 20 mA. The electric current is constant, with a tolerance of ±6% at the output voltage range, $V_{OUT}$ = 4 V to 15 V [30]. Separate activation inputs for each channel allow for PWM brightness control, linear dimming, or individual disconnection of the LED string (Figure 7).

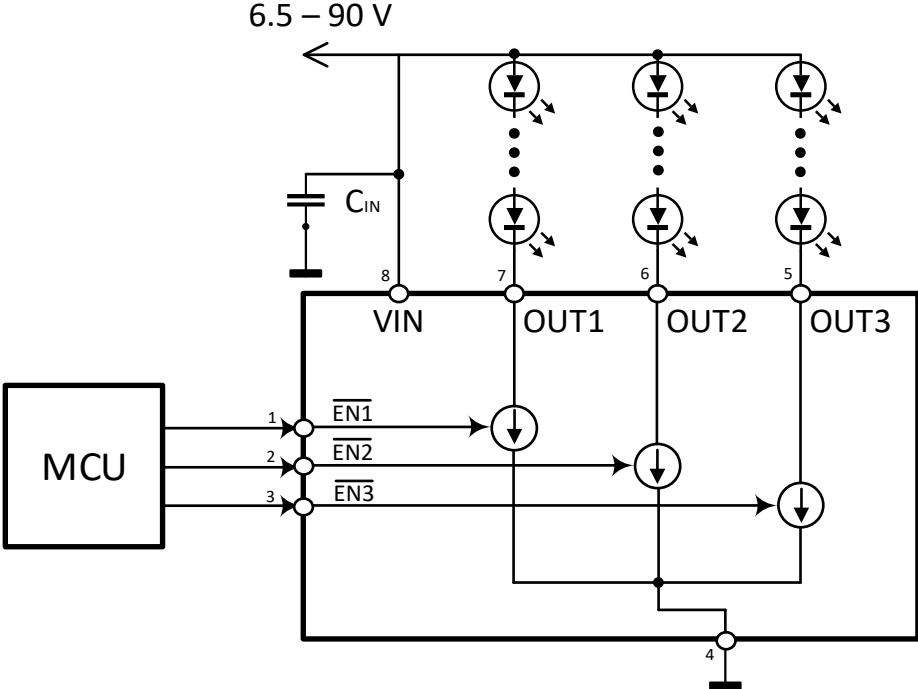

**Figure 7.** Wiring diagram of LED driver, CL320, adapted from [30].

The CL320 also includes an overheat protection circuit that shuts off all three channels at a nominal temperature of 135 °C. Normal operation resumes when the temperature drops by 30 °C. The CL320 is available as an integrated circuit with an SOIC-8 package [30].

Figure 8 shows a sequentially switched-on device with wavelengths of 940 nm, 625 nm, and 525 nm. A total of 60 LEDs correspond to each wavelength. The device is made up of 180 LEDs. Figure 9 shows the application of diffusion filters—Lee 250 Half White Diffusion Filter and Lee 216 Full White Diffusion filter (Panavision, Los Angeles, CA, USA)—to an illumination device to ensure the homogeneous illumination of the examined region of the human cutaneous and subcutaneous tissue. The Lee 250 Half White Diffusion Filter offers medium diffusion used for soft light effects [31]. The Lee 216 Full White Diffusion Filter offers strong diffusion used for soft light effects [31].

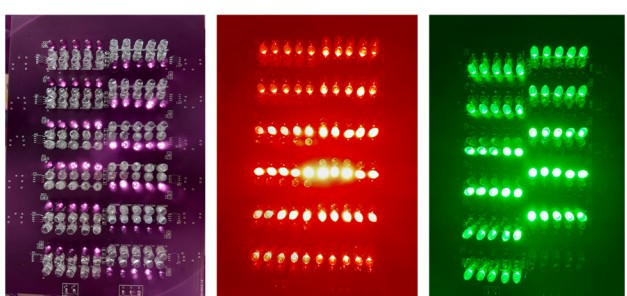

**Figure 8.** Illumination device switched on: infrared, red, and green light.

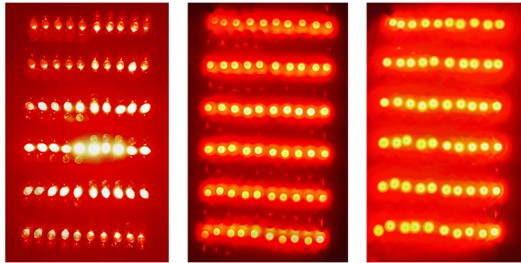

**Figure 9.** Switched-on illumination device diffusion filter differences: **left**—no filter applied, **middle**—Lee 250 Half White Diffusion Filter, **right**—Lee 216 Full White Diffusion Filter.

In addition to comparing the radiant properties of the used LEDs, we decided to introduce their radiance since the technical documentation of LED diodes, OSG59A5111P (525 nm) and OS6YKA5201P (625 nm) (Optosupply, Hong Kong, China), contains data only on luminosity.

$$I = \frac{d\Phi}{d\Omega}$$

$$\Phi_\lambda = k(\lambda) \cdot \Phi_e$$

$$I_v = 683\, \overline{y}(\lambda) \cdot I_e \; (W \cdot sr^{-1})$$

$$I_{e_{525}} = \frac{I_v}{\overline{y}(\lambda)} = \frac{140000 \cdot 10^{-3}}{683 \cdot 0.786} = 260.8\ mW \cdot sr^{-1}$$

$$I_{e_{625}} = \frac{I_v}{\overline{y}(\lambda)} = \frac{120000 \cdot 10^{-3}}{683 \cdot 0.323} = 543.9\ mW \cdot sr^{-1}$$

$$I_{e_{940}} = 550\ mW \cdot sr^{-1}$$

(4)

where I—radiant intensity (W/sr), $\Phi_\lambda$—spectral flux (W/m), $\Phi_e$—radiant flux (W), $I_{e_{525,625,940}}$—spectral intensity (W/sr ·m), $I_v$—spectral intensity (W/sr ·Hz).

### 2.5. Testing of Illumination Device and Camera Setup

When testing the illumination device, which was described in the previous chapter, it is important to mention the camera used, which will be used to capture images at predefined intervals. Our system uses a camera called BFS-U3-28S5M-C USB 3.1 BlackflyS. It is a monochrome CMOS camera that provides a resolution of 1936 × 1464, 2.8 MP, and a 12-bit AD converter [32]. A fast USB 3.1 interface can be used to communicate with the computer. The quantum efficiency of the camera sensor at selected wavelengths can be seen in Table 3.

**Table 3.** Quantum efficiency of the BFS-U3-28S5M-C camera, according to [32].

| Wavelength (nm) | Quantum Efficiency (%) |
| --- | --- |
| 525 | 79% |
| 625 | 70% |
| 940 | 10% |

The set camera exposure time was 8000 µs during testing. The automatic gamma correction was turned off, and the camera gain was set to 5 dB. The camera's AD converter was set to the Mono12p mode with a 10-bit resolution. The camera was controlled by a microcontroller, and thus, operated in slave mode. The sampling frequency was 15 Hz per channel. When testing the device, we focused on the forearm area, where we marked the region of interest, Figure 10. The region of interest size was 3 cm × 2 cm, and its distance from the camera lens and the illumination device, located in the plane with the camera, was 20 cm.

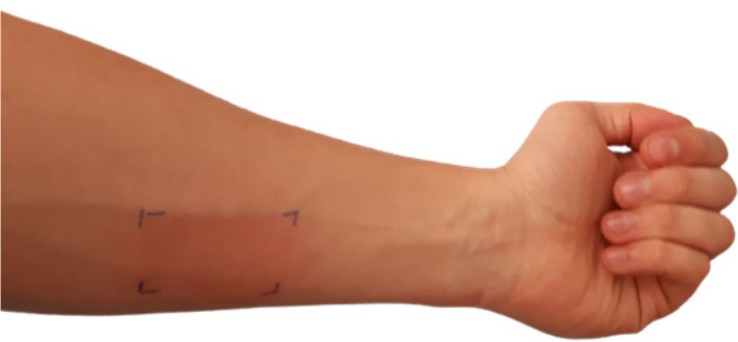

**Figure 10.** Forearm with marked region of interest.

The hand, and thus, the region to be examined, was placed on the foam pad, as shown in Figure 11. The foam pad provides a soft surface for the examined area of the human body. A soft surface is important to avoid unwanted effects on the blood supply to the region of interest. Figure 11 shows the illumination device and measuring setup. Figure 11 shows the foam pad, illumination device, camera lens, power supply, tripod, display, and keyboard.

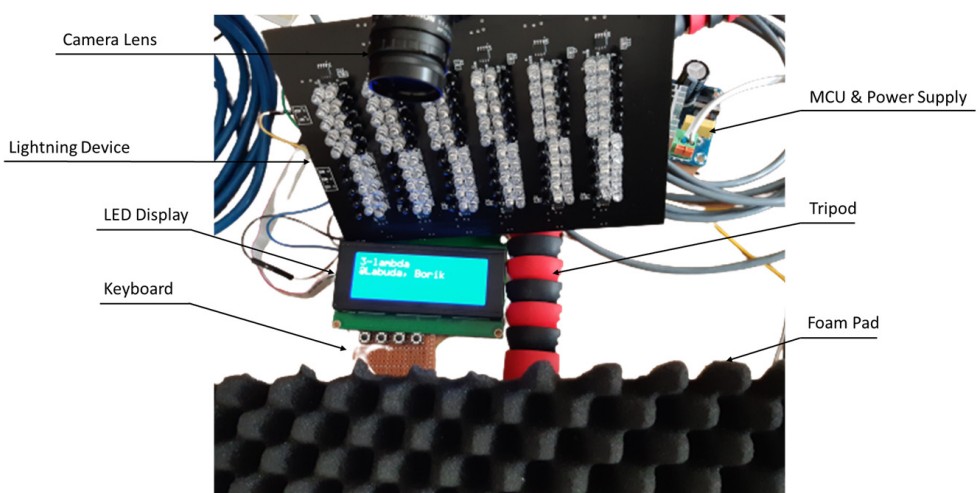

**Figure 11.** Measuring system configuration.

### 2.6. Software Calibration

To compare the DC levels of the PPGI curves, we had to calibrate them in a certain way. We also wanted to compare the maps after calibration and see the effects of the reflectivity and quantum sensitivity of the camera sensor on different wavelengths. The DC components of the PPGI curves are caused by changes in other tissue components, such as venous and capillary blood, bloodless tissue, muscle, etc. [33]. When measuring the PPGI in this section, we thought about the reasons why the individual DC levels of the measured PPGI curves were different. Thus, the curves were not based on the same initial value. We note that this may have been due to the radiant power of the LEDs, the LED driver used with a constant output current value, the sensitivity of the camera for different wavelengths, the reflectivity of the skin, and the absorbance of the individual skin layers. Therefore, we decided to perform the PPGI measurements using white cellulose paper with a density of 80 g/m$^2$ as a reference for the software calibration (Figure 12). An alternative to white paper could be Spectralon [34], which has a defined reflectance of 2% to 99%.

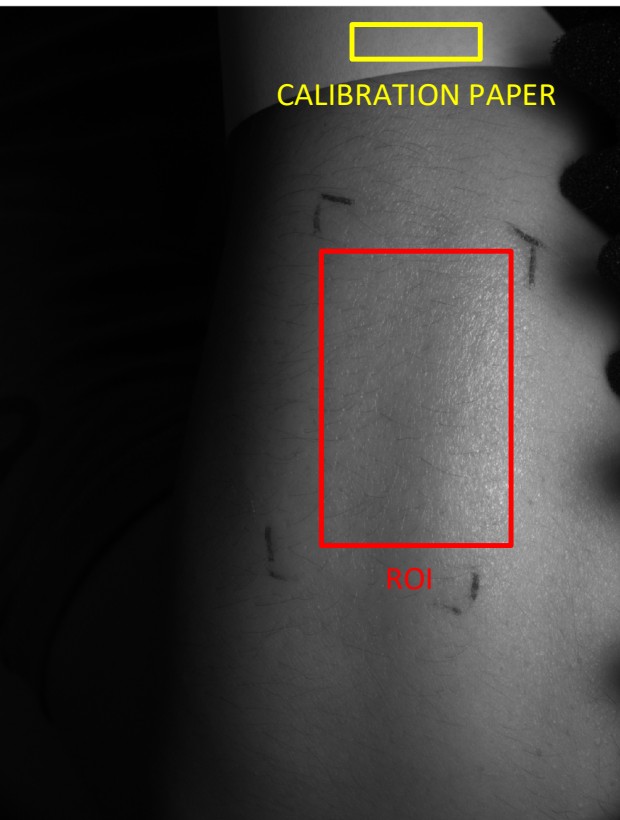

**Figure 12.** Region of interest and white paper for software calibration.

### 3. Results

We tested the illumination device in order to determine the effect of transients during the switching of LEDs on the stability of the designed system. Testing was performed on an oscilloscope DS1102E (Rigol, Beijing, China), measuring transients when switching on and off LEDs of specific wavelengths. We investigated the stability in both the time and voltage domains. Deviation in the time domain can affect the device in terms of system stability in ensuring synchronization between the camera and the illumination device. The short deviation in the voltage range can affect the stability of the illumination power. Deviation in the voltage domain can be a source of noise in the measured signal, and the brightness of the LEDs may be affected too.

#### 3.1. Testing the System Timing

Figure 13 shows the measured time diagram of the whole system, and thus, the switching on and off of the illumination device, including the clock signal (capture signal) for the camera. During this measurement, performed on an oscilloscope, we wanted to verify the time sequence of the system events (camera control, illumination device control) and their synchronism. From the point of view of the time diagram, Figure 13, the duration of the sampling period $T_s$ represents the maximum possible time to switch on the illumination device, and thus, the LEDs of the desired wavelength, and at the same time, capture the image and send it to the computer. $T_{cs}$ is the complete cycle of the sampling period and represents the switching time between all wavelengths of the LEDs to the default state.

$$T_{cs} = \frac{1}{T_s \cdot 3 \text{ channels}} = 22,222 \ \mu s \tag{5}$$

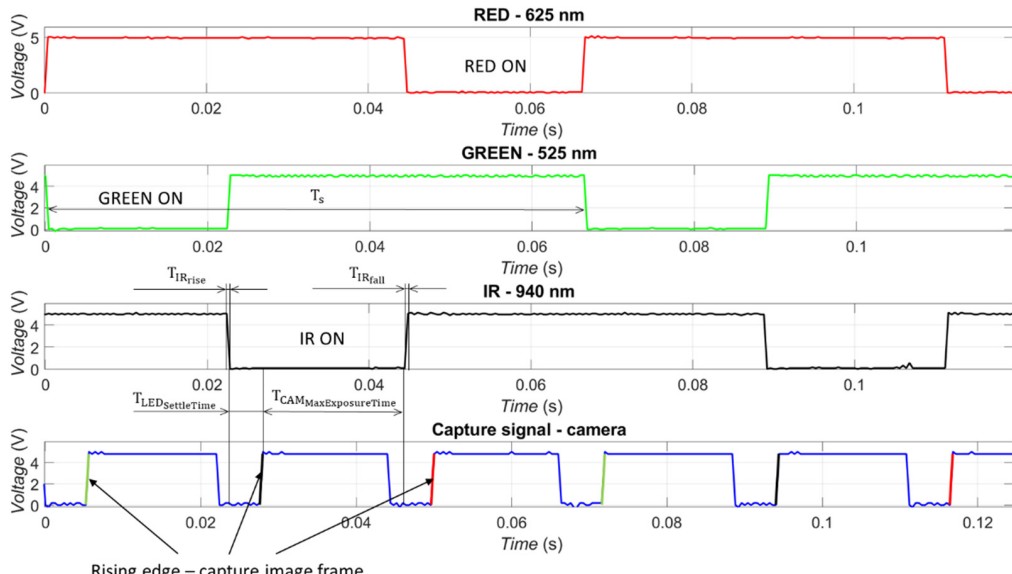

**Figure 13.** Timing diagram of the proposed system, capture signal (blue color), enable signal of LEDs with wavelength of 625 nm (red color), enable signal of LEDs with wavelength of 525 nm (green color), enable signal of LEDs with wavelength of 940 nm (black color).

The rising edge in the camera clock signal represents the beginning of the capture image frame. It is necessary to ensure that the rising edge of the clock signal corresponds to the switched on illumination device with the desired wavelength of the LEDs.

As can be seen in the timing diagram of the illumination device, Figure 13, it is important to correctly select the time required for the LED to settle, $T_{\text{LED}_{\text{SettleTime}}}$, after its activation. When using a sampling frequency of Fs = 15 Hz, the LED settle time can be expressed as:

$$T_{\text{LED}_{\text{SettleTime}}} = \frac{T_{cs}}{4} = \frac{22,222\ \mu s}{4} = 5555\ \mu s \tag{6}$$

Another parameter affecting the functionality of the device is the ideal, $T_{\text{CAM}_{\text{IdealExposureTime}}}$, and maximum, $T_{\text{CAM}_{\text{MaxExposureTime}}}$, camera exposure time, while it is necessary to ensure the following ratio between the mentioned times at all selectable sampling frequencies:

$$T_{\text{CAM}_{\text{IdealExposureTime}}} \leq \frac{2}{3} \cdot T_{\text{CAM}_{\text{MaxExposureTime}}} \tag{7}$$

*3.2. Testing the System Transients*

Subsequently, the transients when switching the LEDs from the active to the inactive state and vice versa were also tested. At the same time, the effects of their switching on the overall time and voltage stability of the device were tested.

Figure 14 represents the switching off of the LEDs with a wavelength of 525 nm and the switching on of the LEDs with a wavelength of 940 nm. Figure 14 shows that the measured transients in the time domain were at the following levels: $T_{625_{\text{rise/fall}}} = 67.60\ \mu s$, $T_{525_{\text{rise/fall}}} = 61.60\ \mu s$, and $T_{940_{\text{rise/fall}}} = 66.00\ \mu s$. In the amplitude domain, the overshoot was at the level of $V_{525,\ 625,940_{\text{overshoot}}} = 0.77$ V. The voltage peak stabilized immediately after stabilizing the deviation in the time domain.

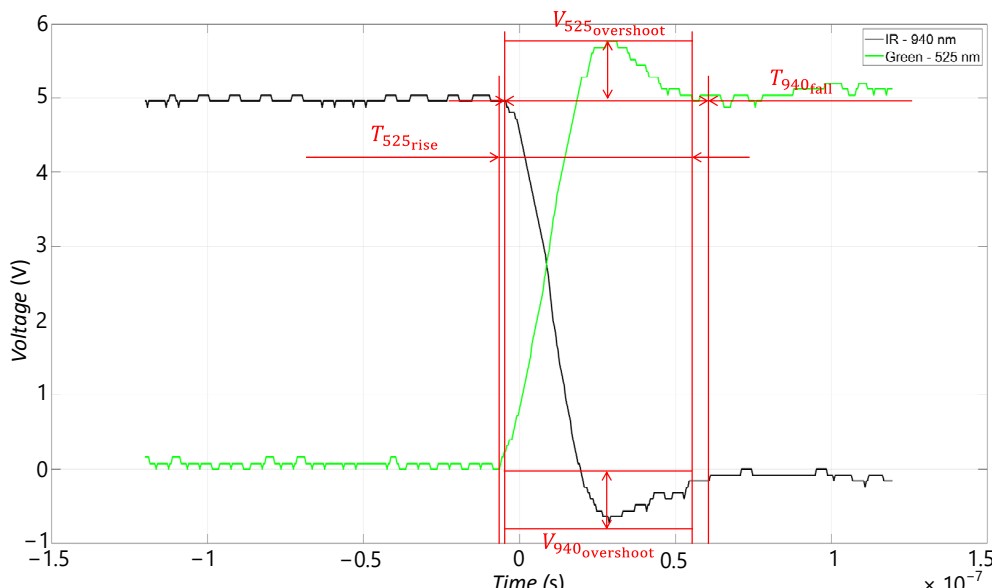

**Figure 14.** Recorded transients when the LEDs with a wavelength of 940 nm were switched on and the LEDs with a wavelength of 525 nm were switched off.

Figure 15 shows the transient when switching off the LEDs with a wavelength of 525 nm. Figure 15 also shows, in the time domain, the interference time of $T_{625_{interference}}$ = 72.4 μs, and in the amplitude domain, the voltage overshoot of $T_{625_{overshoot}}$ = 1.68 V of the LEDs with a wavelength of 625 nm. From this measurement, we can say with certainty that by switching off any LEDs with the desired wavelength, the transient is also transferred to the other LEDs with different wavelengths.

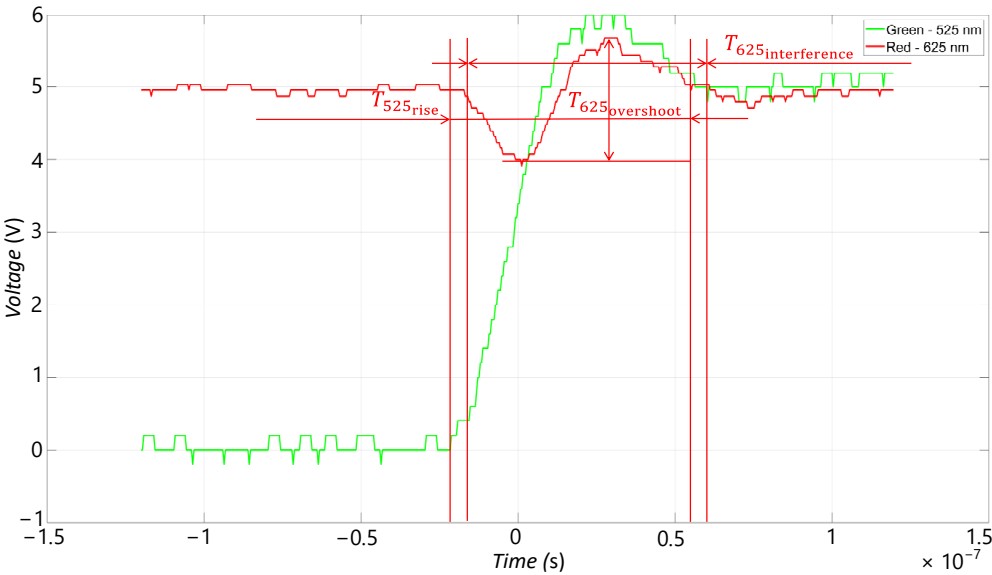

**Figure 15.** Recorded transients when the LEDs with a wavelength of 525 nm were switched off and the LEDs with a wavelength of 625 nm remained switched off.

Figure 16 shows the transient when switching on the LEDs with a wavelength of 625 nm. Figure 16 also shows, in the time domain, the interference time of $T_{525_{interference}}$ = 69.2 μs, and in the amplitude domain, the voltage overshoot of $T_{525_{overshoot}}$ = 0.96 V of the LEDs with a wavelength of 525 nm. We can say from the measurement that by switching on any LEDs with the desired wavelength, the transient is also transferred to the other LEDs with

different wavelengths. The measured parameters mentioned above affected the design process of the entire device in the time domain.

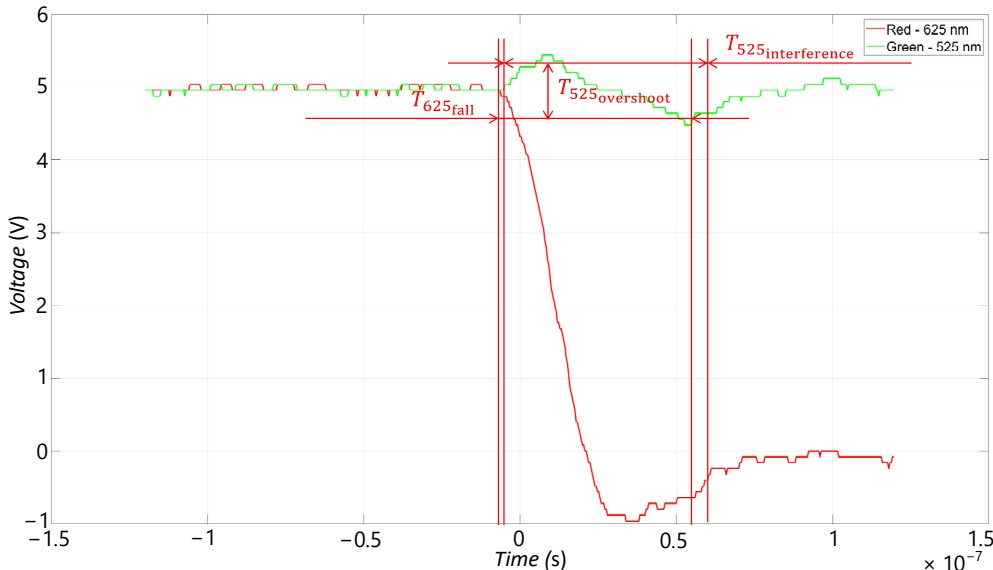

**Figure 16.** Recorded transients when the LEDs with a wavelength of 625 nm were switched on and the LEDs with a wavelength of 525 nm remained switched off.

In Figures 14–16, we can see the three measured model situations from the point of view of the transients that occurred when switching on and off the LEDs of the required wavelengths of the illumination device.

### 3.3. Measuring PPGI Data

The PPG waveform from the region of interest (forearm of the left hand of the measured subject) was measured. Our aim was to capture the AC and DC components of the PPG signal. We used the sequential pulse mode and the 3-λ mode for the measurement. We also applied a vasodilatory cream during the measurement of the region of interest. Finalgon Cream (Boehringer Ingelheim International GmbH, Ingelheim am Rhein, Germany) works by increasing blood circulation to the area of application, hence promoting localized warmth and heat to relieve pain. It is the effect of Finalgon Cream that can be seen on the DC component of the PPG curve. The cream gradually increases the blood supply to the examined area, and thus, increases the rate of light absorption by the tissue, decreasing the magnitude of the resulting PPG signal. Slowly decreasing the DC level over time compared to the initial state can also be called DC drift. The averaged PPG curves from the whole region of interest for the individual wavelengths can be seen in Figure 17a.

The individual waveforms at different wavelengths, Figure 17a, differ in terms of the amplitude level of the signal, which is also related to the sensitivity of the camera to the individual wavelengths of the illumination device and also to the brightness of the selected LEDs that make up the illumination device. At the same time, it is necessary to realize that light with different wavelengths has a different depth of penetration into the tissue, and thus, interacts with different biological structures.

Figure 17b–d shows 5 s details from the measured records, where it is possible to see the PPG waveforms measured at the region of interest. By the test measurement, we wanted to verify the functionality of the illumination device and the entire system when measuring slow and fast changes in the PPG signal. Looking at Figure 17b, it can be seen that the light with a wavelength of 525 nm had the largest magnitude of the AC component of the PPG signal. On the contrary, the light with a wavelength of 940 nm had the smallest magnitude (Figure 17d).

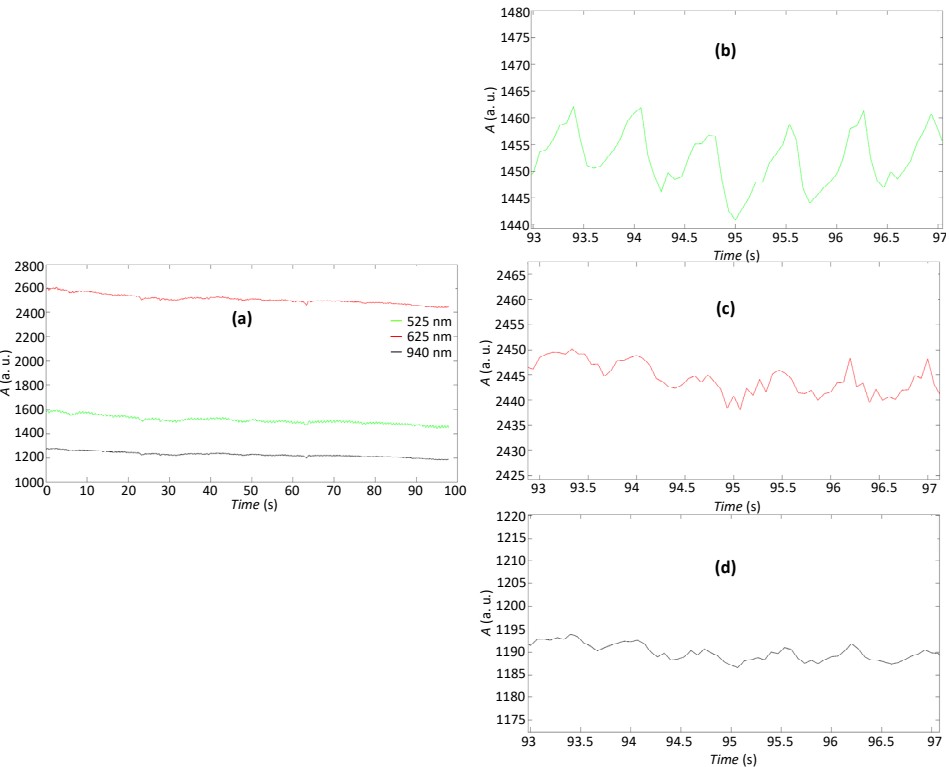

**Figure 17.** Measured PPGI data: (**a**) measured waveforms at three wavelengths, device set in sequential pulse mode; (**b**) a five-second detail of 525 nm waveform; (**c**) a five-second detail of 625 nm waveform; (**d**) a five-second detail of 940 nm waveform.

In Figure 18, it is possible to see PPGI maps created from the measured area (forearm). The size of the kernel when creating the amplitude maps was 15 px × 15 px, and the kernel was moved along the data matrix in steps of three. To extract the amplitudes of the perfusion changes, we used fast Fourier transform (FFT) and tracked the decadic logarithmic maxima in the amplitude spectrum in the range from 0.8 Hz to 2.5 Hz. We can see that the AC components of the 525 nm and 625 nm PPGI waveforms had the highest magnitude levels.

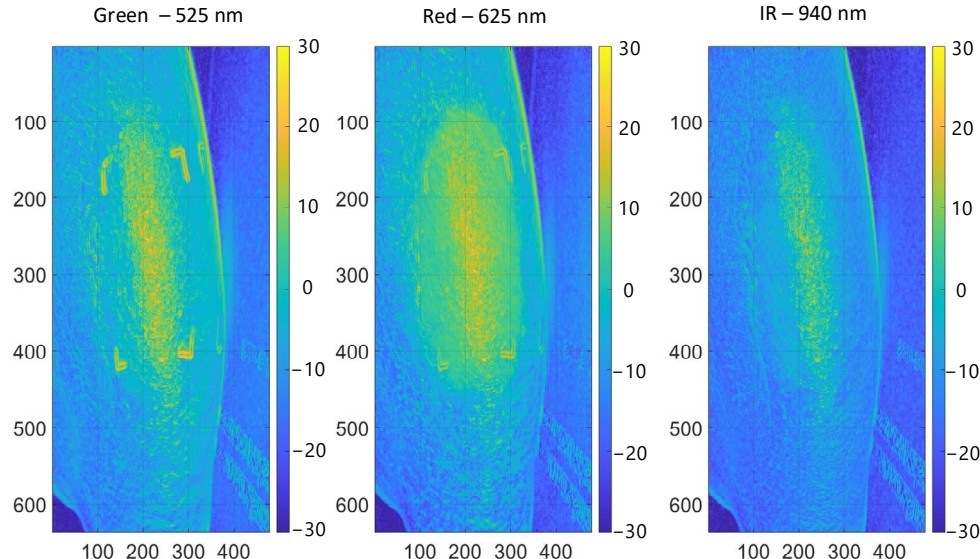

**Figure 18.** Amplitude AC maps, 3-λ mode.

### 3.4. Amplitude Maps after Calibration

We placed the white paper at the same distance from the camera as the measured area of interest. We observed how the DC levels obtained from the area where the paper was located behaved. Figure 19a shows that the maximum DC level on paper was achieved at a wavelength of 525 nm, while the lowest level was reached by radiation at 940 nm. This is due to the mutual influence of the spectral sensitivity of the camera and the radiance power of the LEDs. Therefore, we decided to perform a calibration so that these DC levels could be compared with each other, and so we adjusted the DC level of the red and infrared radiation to the green radiation. Subsequently, we calculated the ratio ($R_{625/525}$) between the DC levels of red ($L_{625_{whitepaper}}$) and green radiation ($L_{525_{whitepaper}}$) and divided the obtained value of red radiation ($L_{625_{skin}}$), carrying out the same for infrared radiation ($L_{940_{skin}}$).

$$R_{625/525} = \frac{L_{625_{whitepaper}}}{L_{525_{whitepaper}}} \tag{8}$$

$$R_{940/525} = \frac{L_{940_{whitepaper}}}{L_{525_{whitepaper}}} \tag{9}$$

$$L_{625_{calibrated\_skin}} = \frac{L_{625_{skin}}}{R_{625/525}} \tag{10}$$

$$L_{940_{calibrated\_skin}} = \frac{L_{940_{skin}}}{R_{940/525}} \tag{11}$$

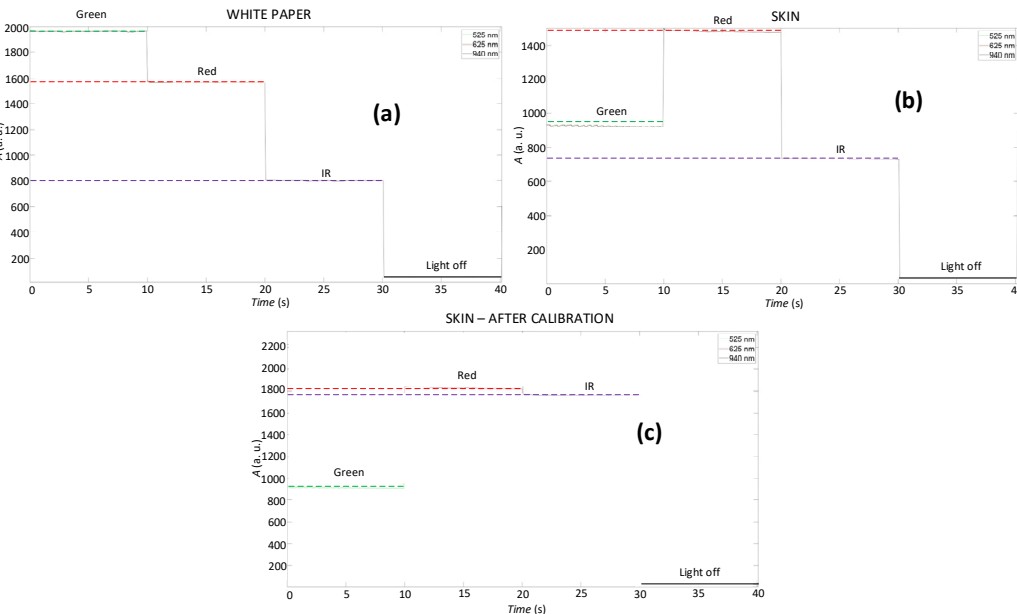

**Figure 19.** Software calibration process: (**a**) magnitude of signals observed from white paper; (**b**) magnitude of signals observed from skin; (**c**) magnitude of signals observed from skin after calibration process.

Thus, we obtained the radiation found on the mentioned white paper at the same initial level, which is a software adjustment of the camera sensitivity and the power of the LED diodes. We also applied this calibration process to the measured region of interest, where the obtained PPG curves can be seen in Figure 19b. If we look at the skin reflectance curve [35], and at the same time, consider the absorption of hemoglobin [36], then the DC component of the green radiation should be the lowest, which also agrees with the data obtained after the normalization for the skin (Figure 19c). If we look at the reflectance

coefficients, the value of the coefficient is 0.3 for green radiation, 0.6 for red radiation, and 0.65 for infrared radiation. If we also consider the sensitivity of the camera sensor to the radiation of the individual wavelengths, then the mentioned data show that the highest DC level should have red radiation ($L_{625_{\text{calibrated skin}}}$), and infrared ($L_{940_{\text{calibrated skin}}}$) and green radiation ($L_{525_{\text{skin}}}$) should have approximately half that of the DC level. We subsequently managed to verify this assumption in Figure 19c.

If we look at Figure 20a,b, we can see that by adjusting the DC level of the PPGI signals, it is possible to compare the created maps from the point of view, for example, of the reflectance of the skin surface. Without the application of calibration, such a comparison would not be possible. According to the created calibrated maps (Figure 20b), the radiation with a wavelength of 625 nm had the greatest reflectance, which also corresponds to the reflectance curve of human skin. On the other hand, the radiation with a wavelength of 525 nm had the lowest reflectivity [35].

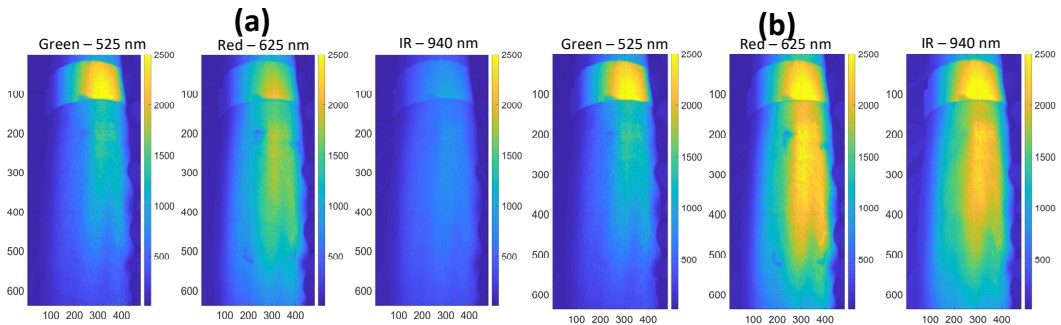

**Figure 20.** Amplitude DC maps, 3-λ mode: (**a**) before calibration; (**b**) after calibration.

In Figure 21a,b we can see that the calibration did affect the AC part of the PPGI signals. This means that the created PPGI maps did not preserve the ratio of the magnitudes of the AC components of the PPGI signals at the selected wavelengths. Therefore, to compare the AC components of the PPGI signals at the selected wavelengths, we had to do so without using the mentioned software calibration or before the applied calibration. The maps were created in the same way as mentioned above when creating the AC PPGI maps without calibration. A vasodilating cream was also used.

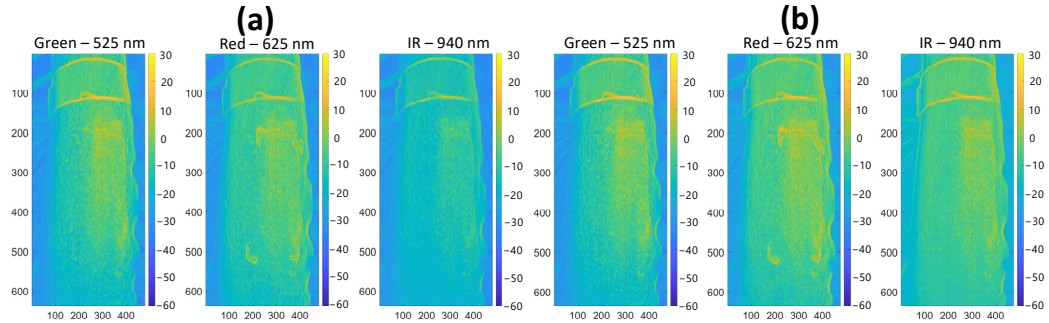

**Figure 21.** Amplitude AC maps, 3-λ mode: (**a**) before calibration; (**b**) after calibration.

## 4. Discussion and Conclusions

The non-contact monitoring of cutaneous perfusion using PPGI carries information not only about heart activity [33] but also about slow changes in the skin and deeper cutaneous tissue blood flow in the region of interest [37].

Our technical solution allows for choosing the desired wavelengths used in the measurement. Thus, it is possible to focus on different skin tissue compartments based on the used light in the optical and near-infrared spectra. The selected wavelength corresponds to

the choice of the examined structure and its anatomical location. The technical solution also allows for changing the intensity and mode of the light source and the parameters of the camera and diffusion filter.

The proposed system, as well as the illumination device, has several limits that do not directly affect the measurement. The first shortcoming is the transients arising from the switching on and off of the illumination device, and thus, the LEDs of the corresponding wavelengths. This shortcoming could be solved in some way by using LEDs with ultra-short switching times and an LED driver with a short on and off time [38,39]. Transients caused by the power supply can be eliminated by using power supplies with high stability when switching the LED load or by designing one's own power supply according to [40–42]. The second shortcoming is related to the camera. Ideally, the image captured by the camera would be processed and sent to a computer immediately. However, because every event of the proposed system, e.g., capturing an image, processing it, and sending it to a computer, takes some time, this can be seen as limiting. This issue can also be solved in a certain way by choosing a suitable camera [43] that shortens the mentioned times.

The advantage of the proposed system is the possibility of choosing the luminosity of the illumination device and the possibility of choosing the specific wavelengths of the LEDs forming the illumination device. The user can select the 1-$\lambda$, 2-$\lambda$, or 3-$\lambda$ mode, as well as the sampling frequency of the image capture. It is important to mention that it is possible to chain the illumination device to achieve higher radiant power and a broader illuminated area.

In focusing on the AC components of the measured PPGI signals (Figure 17), we found that the AC components of 525 nm and 625 nm PPGI waveforms had the highest magnitude levels. This is probably because light with a wavelength of 525 nm penetrates to the depth of the skin where the arterial network is located. It is also important to consider the hemoglobin absorption of radiation [37,44]. The large magnitude of light with a wavelength of 625 nm could also be caused, for example, by the reddening of the skin where the dermal ointment was applied.

The DC components of the measured PPGI signals also depend on the skin reflectance [45] and the camera quantum efficiency [32]. In addition, they were modulated due to the different radiation levels of the selected LEDs. Therefore, it was necessary to perform a software calibration. Software calibration is also important from the point of view of evaluating the measured PPGI data. The measured data can be displayed in amplitude maps, where the individual DC levels at different wavelengths matter for comparison [33].

An option for future work is to add a hardware calibration option, where the brightness of the individual LED panels could be directly controlled depending on the quantum efficiency of the camera and the reflectance of the skin surface under investigation. Other possibilities are adding more wavelengths or using RGB LEDs, where it would be possible to create colored illumination according to the target scenario of the experiment.

In addition, multispectral photoplethysmography imaging opens up the possibility of examining different parts of the skin tissue simultaneously. Thus, this method is promising for monitoring phenomena between different layers or physiological systems, such as the cardiovascular or neuromuscular system.

**Author Contributions:** Conceptualization, S.B.; data curation, S.B.; formal analysis, B.B.; funding acquisition, M.S.; investigation, M.L.; methodology, S.B.; project administration, M.S. and B.B.; resources, M.S.; software, M.L. and S.B.; supervision, S.B.; validation, B.B.; visualization, M.L.; writing—original draft, M.L.; writing—review and editing, S.B. All authors will be informed about each step of the manuscript processing, including submission, revision, revision reminder, etc. via emails from our system or assigned Assistant Editor. All authors have read and agreed to the published version of the manuscript.

**Funding:** This research was funded by the Grant System of the University of Zilina (Project No. O-22-103/0011-03). The APC was funded by University of Zilina: O-22-103/0011-03.

**Institutional Review Board Statement:** The study was conducted in accordance with the Declaration of Helsinki and approved by the Ethics Committee of the Department of Health of the Zilina Self-Governing Region, Slovakia (05128/2022/0Z-02, 2 May 2022).

**Informed Consent Statement:** Informed consent was obtained from all subjects involved in the study. Written informed consent was obtained from the patient(s) to publish this paper.

**Data Availability Statement:** The data underlying the results presented in this paper are not publicly available at this time but may be obtained from the authors upon reasonable request.

**Acknowledgments:** This work was supported by the Grant System of the University of Zilina (Project No. O-22-103/0011-03).

**Conflicts of Interest:** The authors declare no conflict of interest.

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
