# Peer review of "System for Non-Contact and Multispectral Examination of Blood Supply to Cutaneous Tissue"

_electronics, doi:10.3390/electronics11182958_

Round 1

Reviewer 1 Report

The authors presented a measurement system for non-contact examination of human body tissues.

I would suggest adding more references in the introductory section (lines 28-32).

It is preferable to use the point '.' instead of the comma ',' as a decimal separator, please correct it throughout the text.

In order to improve the manuscript readability, I would suggest increasing the labels font size of plots in figures 17 and 19.

In subsection 3.4, the calibration procedure is not so clear (lines 444-448): "we calculated the ratio between the DC level of red and green radiation, and we divided the obtained value of red radiation, and we did the same for infrared radiation." Is it possible to include formulas to better describe the calibration step?

In Figure 19, with the light off the magnitude of the signals is not zero. Have you carried out the measurements in a dark environment? Can an external light (e.g., sun, lamp) influence the measurement results?

Just a final observation. Is it possible to use an RGB led as a light source in order to illuminate the sample with a wider range of colors?

Author Response

Dear reviewer,

First of all, we would like to express our gratitude for your time and effort in reviewing our manuscript. We greatly appreciate your valuable comments and suggestions. Here is our response to your comments, in the attachment. You can also find a manuscript version where we have highlighted all the changes from the original version of our work in color. 

Kind regards,

Authors.

Reviewer 2 Report

The authors presented a  non-contact examination of tissue perfusion using an optical system, which is interesting work. However, the introduction and literature review part needs major improvement. Moreover, the method part should be coherent and organized to make it easy for the reader to follow. Some of the figures should be moved to supplementary materials which are less useful for presentation. Discussion section should highlight the findings of experiments clearly in sub-sections. Paper should end with a conclusion and take home information. 

Author Response

(The authors gave the same response as above.)

Round 2

Reviewer 2 Report

Authors have improved the article in the revised version. However, there are some issues still there. 

Fig 1 is poor. It should be redrawn.

Some of the figures can be moved to supplementary materials.

Fig 10 is not meaningful and clear

Same is true for Fig 11

Lot of equations are not referenced and numbered